# Distribution of *Aspergillus* Fungi and Recent Aflatoxin Reports, Health Risks, and Advances in Developments of Biological Mitigation Strategies in China

**DOI:** 10.3390/toxins13100678

**Published:** 2021-09-24

**Authors:** Firew Tafesse Mamo, Birhan Addisie Abate, Yougquan Zheng, Chengrong Nie, Mingjun He, Yang Liu

**Affiliations:** 1School of Food Science and Engineering, Food Safety Research Centre, Foshan University, Foshan 528231, China; niecr@126.com (C.N.); hemingjun1996@foxmail.com (M.H.); 2Ethiopian Biotechnology Institute, Addis Ababa 5954, Ethiopia; birhanaddisie@gmail.com; 3State Key Laboratory for Biology of Plant Diseases and Insect Pests, Institute of Plant Protection, Chinese Academy of Agricultural Sciences, Beijing 100193, China; yqzheng@ippcaas.cn

**Keywords:** aflatoxins, *Aspergillus flavus*, occurrence, atoxigenic strains, biocontrol, China, hepatocellular carcinoma

## Abstract

Aflatoxins (AFs) are secondary metabolites that represent serious threats to human and animal health. They are mainly produced by strains of the saprophytic fungus *Aspergillus flavus*, which are abundantly distributed across agricultural commodities. AF contamination is receiving increasing attention by researchers, food producers, and policy makers in China, and several interesting review papers have been published, that mainly focused on occurrences of AFs in agricultural commodities in China. The goal of this review is to provide a wider scale and up-to-date overview of AF occurrences in different agricultural products and of the distribution of *A. flavus* across different food and feed categories and in Chinese traditional herbal medicines in China, for the period 2000–2020. We also highlight the health impacts of chronic dietary AF exposure, the recent advances in biological AF mitigation strategies in China, and recent Chinese AF standards.

## 1. Introduction

Aflatoxins (AFs) are secondary metabolites produced by members of three distinctive sections of the genus *Aspergillus*: section *Flavi*, section *Ochraceorosei*, and section *Nidulantes* [1]. Members of section *Flavi* are the most common and widespread producers of AFs. The most commonly known AF producing *Aspergillus* fungi are *A. flavus* and *A. parasiticus. A. parasiticus* appears to be more adapted to a soil environment, being prominent in peanuts, whereas *A. flavus* seems adapted to the aerial and foliar environment, being dominant in corn, cottonseed, and tree nuts. Thus, it is known to be the most frequently encountered producer of AFs in agricultural products because of its widespread distribution [2,3].

Plant debris, decaying wood, animal silage, dead insects, and animal carcasses are the main organic nutrient sources of *A. flavus* as this fungus is saprophytic in nature [4]. Although it occurs predominantly in aerial and foliar environments [5], it can even reside on human organs. For example, recent studies reported its presence in tracheal aspirates of patients infected with COVID-19 [6,7].

*Aspergillus* fungi can grow and proliferate almost everywhere in the world under variable climatic conditions that range from arid to tropical moist to temperate [8,9]. *A. flavus* requires temperatures ranging from 25 °C to 33 °C and water activity of >0.98 for active growth [10]. However, it can still grow at temperatures between 30 °C and 40 °C [11]. Thus, it occurs in all major cereal, peanut, tree nut, and cotton seed growing areas that experience high temperatures [5].

*Aspergillus* species produce different types of AFs, including the potent parent toxins AFB1, AFB2, AFG1, and AFG2 under natural conditions. Moreover, they generate other metabolites (AFB2_a_, AFG2_a_, AFGM1, AGM2, AFM1, aflatoxicol (R_0_), parasiticol (B3), and aspertoxin) [12]. Crop soils are the primary source of *A. flavus*, which means that important food crops can be invaded in the field and subsequent AF contamination can occur when the plants are under stress due to factors such as high soil and/or air temperature, high relative humidity, drought, or insect attacks [13,14]. AFs are generally carcinogenic [15,16], can suppress immunity [17,18], and can lead to growth impairment in children [19,20] and AFB_1_ is the most potent and frequently occurring AF [21]. *A. flavus* strains are also known to produce cyclopiazonic acid (CPA) [22], which is an indole tetramic acid that was originally discovered in peanuts as a fungal metabolite [23]. AFs and CPA commonly occur as co-contaminants and result in substantial economic losses [24,25].

According to a report published by the International Agency for Research on Cancer (IARC), 500 million people in Asia, Latin America, and sub-Saharan Africa are exposed to AFs at levels that substantially increase mortality and morbidity [26]. Mycotoxins, particularly AFs, may play a causative role in 4.6–28.2% of all global hepatocellular carcinoma (HCC) cases [15]. Globally, food safety is regularly compromised by the presence of major mycotoxins, including AFs [27]. AFs have been reported to be causative agents for increased healthcare costs, reduced livestock production, disposal costs of contaminated foods and feeds, pre- and postharvest crop losses, and research investment [28]. Countries have also been forced to invest huge amounts of capital into regulatory programs aimed at reducing AF levels in end products [29].

Over 100 nations have established maximum tolerable levels for AFs in food [30,31], and several nations have set standards for AFM_1_. The most recent list of AF regulations on a nation-by-nation basis was published by the FAO (2003). 

The most stringent aflatoxin limits are set in the European Union under commission regulation (EU) No. 165/2010 in diversified food categories like peanut, dried fruits, cereals, etc. For example, the upper limits for AFB_1_ and total AFs (B_1_ + B_2_ + G_1_ + G_2_) in peanuts are 2 µg/kg and 4 µg/kg, respectively. Different countries have also established different standard ranges for peanut. In China the upper limit is 20 µg/kg for total AFB1 [30], and the USA and Canada adopted a similar total AF limit (15 µg/kg) [32]. In Asia, Singapore has the lowest total aflatoxin limit (5 µg/kg), whereas India has a relatively less stringent standard for AFB1 in peanut (30 µg/kg). 

The populations of *A. flavus* can be subdivided into different groups based on sclerotia size (L-strain > 400 μm in diameter and S-strain < 400 μm) [33]. Depending on their geographic origin, some S-strains produce both AFBs (AFB_1_, AFB_2_) and AFGs (AFG_1_, AFG_2_), whereas others produce only AFBs [34]. Other isolates with abundant small sclerotia (diameter < 400 µm) are classified as strain SBG [34]. Within the *A. flavus* population, some strains may produce different ranges of AFs and are referred to as toxigenic, whereas the rest may not secrete any toxins and are referred to as atoxigenic [35,36,37,38]. Most of the atoxigenic isolates of *A. flavus* belong to L-strains. 

Asia and Africa are the continents most affected by AF contamination. China is among the countries with a high prevalence of AFs in different agricultural products targeted for both domestic consumption and export. Studies have reported higher levels of AF contamination in crops from the southern part of China (such as Guangdong Province) compared with other regions [32,39]. 

Researchers have conducted a number of studies to assess the occurrence of AFs and the distribution of fungi in agricultural products [37,38,40].The occurrence of AFs in agricultural commodities has been fairly well characterized because of the importance of understanding contamination of the food supply. However, analyzing the dynamics of *A. flavus* distribution with respect to specific geographical locations, crop fields, and different crop varieties is equally important. Precise mapping and a wider understanding of the distribution of *Aspergillus* species as well as their AF production capacity in different geographical locations, are crucial for designing appropriate mitigation strategies. Initiatives to explore the population distribution and to characterize *A. flavus* strains have been undertaken in different countries of the world, including the USA, Nigeria, Italy, Argentina, Iran, and Thailand. In these studies, researchers aimed to differentiate toxigenic and atoxigenic strains among the total *A. flavus* population using different polyphasic sequential approaches, such as morphological, microbiological, and biochemical techniques [4,35,36,37,38,40,41,42,43]. Furthermore, molecular characterizations were conducted by targeting all or some of the important genes among the AF biosynthetic gene cluster. 

Recently, interest in documenting the distribution of *A. flavus* across China has increased because of the potential for using isolates of atoxigenic strains to reduce AF contamination [44,45,46]. The distribution of *A. flavus* in different agro-ecological zones has been reported [40,47], as have the molecular characteristics of potential atoxigenic *A. flavus* strains [40,45,48]. Moreover, in China different research teams are studying AF biological control in the laboratory and in the field. The goal of this review is to compile this information to document what has been done to date so that it can be used as a data source for future studies. We present an update of information about AF contamination in different food commodities, AF distribution, molecular characteristics of the main AF-producing molds, AF-related health risks, and recent developments in AF biocontrol as mitigation strategies in China. We also discuss future perspectives.

## 2. Reports of AF Distribution in Different Commodities in China

Mycotoxin contamination is a prevailing problem in China, and it is complicated by China’s export of several agricultural products to the EU and the USA. Several studies have reported the presence of AFs in agricultural products such as cereals, peanuts, spices, milk, and animal feeds that originate from China. Several Chinese medicines have also been found to contain mycotoxigenic fungi as well as their toxins. AF contamination reported in different commodities from China from 2001 to 2020 is summarized in the tables below. 

### 2.1. Cereals Crops 

Cereals, mainly, maize, sorghum, wheat, and rice, are the most AF-vulnerable crops. A number of authors in different countries, mainly Africa and Asia, have reported AF contamination of these crops. Table 1 lists AF contamination reported in different cereal and cereal-based food products. The main food items reported to contain AFs were maize, wheat, oats, and rice and their processed products, which were collected from almost all parts of the country. The majority of the food samples tested positive for AFs. The prevalence and incidence of AF contamination are generally higher in maize. Wang et al. [49] found that 76.7% of maize samples collected from Zhuqing Village, Fusui, and Guangxi Province were contaminated with AFB1 (0.4–128.130 µg/kg), and an alarming 30% of AFB1-positive samples contained levels beyond the maximum national limit (20 µg/kg) in maize. Similarly, 45% of maize samples collected from Chongzuo County and Guilin suburbs in the Guangxi Autonomous Region were contaminated with a wide range of AFB1 levels (9–2496 µg/kg, average 460 ± 732 µg/kg), and 76% of the positive samples contained levels higher than the Chinese limit [50]. In an investigation conducted in the main maize producing region (Yangtze Delta) of China, Li et al. (2014) [51] found that 14.5% of samples (76 in total) were positive for AFB1 (1.0–32.2 µg/kg, average 6.6 µg/kg), and 4% and 9.2% of the positive samples surpassed the national maximum limit and the EU maximum limit, respectively. 

The concentrations of AFs in rice are much lower than those found in other cereal foods, but the incidence of AF-positive rice samples is alarmingly higher [52,54]. China was the world’s leading rice producer and consumer in 2017/2018 [60], and rice is consumed daily in every Chinese household. Consequently, long-term exposure of even insignificant levels of AFs may pose a serious health risk.

Generally, the level of contamination in most examined foods is much lower than the national maximum AF limit of 20 µg/kg, but the incidence of AF contamination is very high. In addition, the overall level of contamination in the southern part of China is higher than that in the northern region, possibly due to the hot and humid climatic conditions in the south. Long-term exposure to low levels of AFs could have negative health impacts on consumers. Therefore, a system of AF mitigation is needed, and it should target prevention as well as adherence to strict national standards and guidelines. Such tactics can be effective, as recent socioeconomic changes, education, and the application of postharvest preventive mitigation strategies have reduced AF contaminations of maize [58].

### 2.2. Peanuts, Pine Nuts, Nuts, Oils, and Other Oil Products 

Table 2 lists a variety of AF-contaminated commodities, such as peanuts, walnuts, pine nuts, peanut butter, sesame paste, peanut oil, vegetable oil, sunflower oil, fish oil, and maize oil, that were collected from every part of China.

Peanuts are among the most important oil seeds produced by different countries around the world. Global peanut production reached approximately 46.78 million metric tons in the 2018/2019 growing season; China was the world’s largest producer, contributing 17.33 million metric tons, followed by India, Nigeria, and the USA with 4.72, 4.42, and 2.49 million metric tons, respectively [65]. Five provinces of China (Guangdong, Shandong, Henan, Hebie, and Jiangsu) contribute 70% of the country’s production [66]. Most of the peanut production is located in the southern and southeastern regions of China, which are characterized by the relatively highhumidity and temperature favorable for *A. flavus* growth and AF contamination at the preharvest stage [67]. Several studies have reported higher levels of AF contamination in peanuts from the southern part of China, such as Guangdong Province [32,39]. 

China is the main exporter of peanut and peanut products to EU countries. 

In a survey of peanut butters and sesame pastes purchased from retail markets in Beijing, Shanghai, Changchun, Chengdu, Shijiazhuang, and Zhengzhou, 82% (out of 50) peanut butters and 37% (out of 50) sesame pastes were AFB1 positive, with values of 0.77–70.64 µg/kg and 0.54–56.89 µg/kg, respectively [63]. Moreover, 19% of the sesame pastes and 2% of the peanut butters contained levels that exceeded the Chinese maximum limit for AFB1 of 5 µg/kg. The EU maximum total AF limits and AFB1 limits were surpassed by 39% and 37% of the peanut butters, respectively. Similarly, 24% and 32% of the sesame pastes contained levels beyond the EU maximum limits of total AFs and AFB1, respectively. These findings highlight the potential health and economic risks of these foods, as China is the largest sesame paste consumer and the EU is one of the main importers of peanuts and peanut products from China.

In a two-year survey of more than 1000 peanut samples collected from 12 provinces (potential agricultural areas) of China, one-quarter of the samples tested positive for AFB1 (0.01–720 µg/kg, average 2.13 µg/kg) [61]. Furthermore, 1% and 3.7% of the peanut samples contained AFB1 levels above the Chinese regulation and the EU maximum limit, respectively. Additionally, AFB2, AFG1, and AFG2 were detected in peanut samples. Although these are known to be carcinogenic, they are not yet regulated in China. In the same study, different oil products tested positive for AFs (Table 2), but in general the AFs were present at safe levels and the incidence was minimal. However, Zhang et al. (2020) [58] conducted a survey of 1854 commodities, including rice, wheat, maize, nuts, vegetable oils, and homemade peanut oils, collected from 11 districts in Guangzhou and found that 75% of 96 homemade peanut oil samples contained AFB1 (0.26–283.0 μg/kg, average 38.74 ± 47.45 μg/kg). The average value was seven times higher than the maximum permissible limit set by the Chinese government (5 μg/kg). 

In another survey of cereals, peanuts, and oils collected from the Yangtze Delta region of China, 14.5% of the samples tested positive for AFs (Li et al., 2014). The maximum concentration of total AFs was 35.0 µg/kg in peanut butter, and the AFB1 level was 32.2 µg/kg. The peanut butter sample with the maximum AF contamination also contained the highest concentration of AFB1 (32.2 µg/kg) [51]. Moreover, 4% of the samples surpassed the maximum tolerable limit set by the Chinese government. 

Sun et al. (2011) [54] collected 209 maize, rice, wheat, peanut, and plant oil samples from Shandong, Jiangsu, and the Guangxi Zhuang Autonomous Region, and all of them tested positive for AFB1. Among the commodities tested, plant oil from Fusui County, Guangxi contained the highest amount (0.5–114.4 μg/kg, median 52.3 μg/kg), and it was much higher than the Chinese maximum limit (10 μg/kg) (GB2761-2005). Additionally, Fusui County has elevated numbers of HCC cases, and plant oil was identified as the main source of AF exposure. 

Reports suggest that most of the AF exposures that correlate with HCC cases are due to raw peanuts or peanut-based foods or cooking oils. For example, Wang et al. (2001) [49] reported that peanut oil was the major source of AFB1 in Fusui County, Guangxi Province, and this locale has a high number of HCC cases. According to a survey of 12 agricultural provinces in the eastern part of China, daily AFB1 intake from raw peanuts was estimated to be 0.11–5.66 ng/kg bw/day, and the population risk of developing HCC was 0.003–0.17 cases/year/100,000. The risk from peanut oil was 10 times higher (0.84–68.8 ng/kg bw/day; 0.03–2.06 cancer cases/year/100,000) [61].

Regulatory agencies limit the maximum tolerable limit of AF in peanuts. The upper limit for AFB1 in peanuts is 2 µg/kg and 4 µg/kg for total AFs in the European Union, whereas China has a tolerance of 20 µg/kg for total AF [30] and AFB1 in peanut and peanut products. Further, the country has set a maximum AFB1 limit of 10 µg/kg for vegetable oil for domestic use (other than peanut and maize) [68]. 

Stringent global AF regulations are influencing the country’s export market. Peanut AF contamination is a challenge for peanut export from China to the EU. A report posted 3 May 2018 on the website Food Navigator [69] stated that peanuts exported to the EU contained AF levels above the EU Commission Regulation (EC) 1881/2006. The article also stated that after repeatedly finding AFs in peanuts from China, EU regulation 884/2014 imposed special conditions on the import of in shell and shelled peanuts as well as peanut butter. According to Rapid Alert System for Food and Feed (RASFF) reports, only between January 2020 and June 2021, around 21 notifications have been released due to excessive aflatoxin detection from peanut and peanut products that originated from China [70] (RASFF, 2021). These notifications were from different EU member states such as Belgium, the Netherlands, Spain, the United Kingdom, the Czech Republic, Italy, Bulgaria, France and Croatia. 

AF contamination of peanuts is also a serious challenge for the export sector of several African and other Asian countries.

### 2.3. Chinese Herbal Medicines (CHMs), Spices, Tea, Fruits, and Vegetables

CHMs play an important role in treating diseases in China at a level almost equal to that of modern drugs. China is the major country involved in the production, consumption, and export of CHMs, and the demand in recent years has significantly increased. Annual consumption of CHMs in China exceeds 400,000 tons [71], and over 100 million Europeans currently use Traditional and Complementary Medicine for health care [72]. Due to the large quantities of CHMs being used in foreign countries, safety of these products is receiving a lot of attention, as these products are susceptible to mycotoxigenic fungal contamination during production, processing, transportation, and storage. Toxigenic fungi species that are generated from soil or plants themselves can result in contamination of herbal medicines. Therefore, different countries in the world have set maximum limits of AFs in herbal medicines. Generally, the current legal limit for AFB1 in medicinal herbs ranges between 2 and 10 µg/kg, and the limit for total AFs (combined AFB1, AFB2, AFG1, and AFG2) ranges from 4 to 20 µg/kg [73]. In China, the maximum limits in herbs are 5 µg/kg and 10 μg/kg, respectively [74]. CHMs that have maximum limits for AFs in China are as follows: *Jujubae Fructus, Hirudo, Pheretima, Myristicae Semen, Scorpio, Cassiae Semen, Hordei Fructus Germinatus, Polygalae Radix, Citri Reticulatae Pericarpium, Qutisqualis Fructus, Platycladi Semen, Sterculiae Lychnophorae Semen, Nelumbinis Semen, Persicae Semen, Scolopendra, Arecae Semen, Ziziphi Spinosae Semen, Bombyx Batryticatus*, and *Coicis Semen* [74]. EU countries have also established stringent standards, such as commission regulation (EC) No. 165/2010, which sets the maximum limits of AFB1 for dried fruits and spices in the range of 2 to 5 μg/kg and total AFs between 4 and 10 μg/kg. Table 3 summarizes updated data for AF contamination in herbal medicines, spices, fruits, and vegetables in China. 

### 2.4. Animal Feed and Dairy Products 

The aflatoxins were discovered in the early 1960s, when they were identified as causative agents of “turkey X” disease, due to peanut-based feeds that originated from South America [82]. Subsequently, several research studies have reported that long-term exposure of animals to sub acutely toxic levels of AFs can cause several animal health implications, such as feed refusal, increased disease susceptibility, weight loss, inferior egg shells, reduced milk yield, defects of carcass quality, reproduction defects [83,84], and other severe impacts, such as liver lesions/tumors, hepatotoxicity, nephrotoxicity, gastroenteritis, and teratogenicity [85,86]. 

The presence of microscopic fungi affects the quality of feeds, their organoleptic attributes, and nutritional quality. In addition to their negative impact on nutritional and organoleptic properties, molds can synthesize different mycotoxins. Like any food commodities, unless properly harvested, dried, processed and stored, any feed types can also be suitable for fungal proliferation as well as aflatoxin contamination [87].

Additionally, feeding animals with contaminated feeds leads to the transfer of AFs into the milk in the form of AFMs. For instance, about 1–3% of AFB1 present in feedstuffs appears in milk as AFM1 [88]. The presence of AFMs in milk leads to the contamination of dairy products, as these AFs are not eliminated by the typical processing used by food industries or by food cooking [89]. AF contamination in cow’s milk poses a risk to humans because it is an important foodstuff for both children and adults. Therefore, the presence of AFM1 in milk and milk products is considered to be undesirable [89,90]. 

Table 4 lists AFs that were detected in feeds and dairy products collected from different parts of China and shows that the prevalence of AFs in both feedstuffs and dairy products is high. However, the levels of most of these toxins are far below China’s maximum limits for feeds and dairy products. 

## 3. AF Detection in China

Dietary exposure to mycotoxins, even at low levels, has been confirmed to be very dangerous for the consumer’s health. Therefore, the demand for sensitive analysis and quantification methods is high [99]. Several mycotoxins detection methods are currently available, and modern and efficient techniques are being developed continuously. High-performance liquid chromatography (HPLC) is known for its sensitivity and accuracy for detecting several mycotoxins. In recent years, a more sensitive and faster method called ultra-performance liquid chromatography (UPLC) was developed [100]. Currently, chromatographic methods combined with mass spectrometry (LC/MS) are also being used to detect and quantify mycotoxins [99]. Commercially available enzyme-linked immunosorbent assays (ELISAs) with a narrow detection range and lateral flow immunoassay (LFIA) kits also provide a relatively easy method for quantifying mycotoxins [101]. In China, as shown in Table 1, Table 2, Table 3 and Table 4 several modern and sensitive methods including thin layer chromatography, ELISA, LFIA, HPLC, LC/MS, LC-MS-MS, and UPLC are employed for mycotoxin detection and quantification. The detection limits (sensitivity) of each method with respect to different commodities are shown in Table 1, Table 2, Table 3 and Table 4. 

Tumukunde et al. (2020) [102] reviewed AF detection techniques commonly used in China. In addition, rapid diagnostic methods that are either immunoassay- or biosensor-based are also being utilized in China. These techniques are important for reducing AF risks, as they are fast, easy to use, and inexpensive. They are also known for their reproducibility, stability, accuracy, and portability for on-site testing. Recent advances in these techniques in China have been described by a number of authors [103,104,105].

## 4. Health Impacts of AFs in China

Human dietary AF exposure can result in either chronic or acute health risks depending on the extent of exposure. In one remarkable case, 125 Kenyans died between 2004 and 2006 due to acute liver failure after consumption of homegrown maize containing high levels (up to 4400 ppb) of AFs [106,107,108]. Stunted growth, immunosuppressive effects, and cancer are also chronic health complications of AF consumption [17,18,108]. Among the AFs, AFB1 is known to cause liver cancer and, in synergism with the hepatitis B virus (HBV), to increase the possibility of developing chronic liver disease (CLD) or HCC [109,110]. 

AF parent molecules (like AFB1) are relatively harmless, but the electrophilic intermediates AFBO (B1-8,9-epoxide) that are generated at the predominant AF metabolization site are mutagenic and carcinogenic [111,112]. The major human cytochrome p450 enzymes are responsible for converting AFB1 into two reactive 8,9-epoxide stereoisomers (exo and endo) [108]. Exo-isomers are more toxic and cause the AFB1 to exhibit genotoxic characteristics [113]. The exo-8,9-epoxide has a high binding affinity for DNA, forming AF-DNA adducts, which primarily exist as 8,9-dihydroxy-8-(N7) guanyl-9-hydroxy-AFB1 (AFB1-N7-Gua) adducts. These adducts are primarily responsible for the genotoxic, mutagenic, and carcinogenic properties of AFB1 [21]. 

Human dietary AF exposure is usually measured using several biomarkers. Biomarkers help to assess AF exposure to more accurately reflect individual intake of AFs, and they are measured in urine or blood serum. The AF albumin adduct (AF-alb) in serum is a valuable biomarker for CLD and HCC due to long-term high AF exposure [114,115,116]. The most commonly used biomarkers are the urinary AFB1-N7-Gua adduct, which is a product of DNA damage, and the metabolites of AFM1 in urine or milk (AFP1, AFB1, AFQ1, AFP1, AFB-N-acetyl-L-cysteine (AFB1-mercapturic acid)) [49,117,118]. 

In China, studies of AFs and HCC risks have been conducted for more than three decades, with the earliest report dating back to 1989 [119]. That study confirmed the roles of the HBV virus and AFB1 in the rate of primary hepatocellular carcinoma (PHC) in southern Guangxi, China, which was the most PHC-prevalent region in the world. A number of other researchers have reported elevated AF exposure and higher incidences of HCC in different cities/counties of China in the provinces/municipalities of Jiangsu, Guangxi, Guangdong, Shanghai, and Taiwan [49,55,120,121], as regions in the southern parts of China are prone to AF contamination because their humid and warm climate is suitable for the growth and proliferation aflatoxigenic fungi [32,39]. 

Li et al. (2021) [50] reported a positive correlation between HCC mortality and AFB1 dietary exposure from maize and peanut oil in Guangxi. In the early 1990s several studies reported the incidence of HCC cases in Shanghai. Ross et al. (1992) [117] reported the correlation between serum hepatitis B surface antigen (HBsAg) positivity and AF exposure as a risk for developing liver cancer in Shanghai, based on urinary AF metabolite levels (P and M) and DNA-adducts. They also attributed up to 50% of liver cancer cases in Shanghai to AF exposure. In another study conducted in Shanghai, Qian et al. (1994) [118] found a strong correlation between the presence of urinary AFs (B1, M1, P1, and AFB1-N7-Gua), HBsAg positivity, and HCC risk. In this study, AF biomarkers were found in 90% of 55 HCC-positive cases.

Guangxi Province is a well-studied region of China due to its high rate of HCC morbidity, mortality, and AF exposure. Wang et al. (2001) [49] conducted a study in Zhuqing Village, Fusui County, Gangxi, aimed at determining the correlation between AF exposure, chronic HBV, and HCC cases. In this study, AFB1 contents of the major food items in the area were evaluated, and it was detected in 76.7% of maize (range 0.4–128.1 ppb), 66.7% of cooking oil (range 0.1–52.5 ppb), and 23.3% of rice (range 0.3–2.0 ppb). The mean levels of serum AFB1-albumin adducts in 29 identified HCC groups were >1.2 pmol/mg of albumin at both the beginning and end of the study period, and urinary AFB1 metabolites were detected in 88.9% of samples (range 0.9–3569.7 ng/24 h urine). The study also concluded that HCC accounted for 64% of the total cancer cases in the area. 

Li et al. (2001) [50] conducted a comparative study of dietary AF exposure in Guangxi. They collected 20 maize samples from 20 farmers in Chongzuo County, which is a high PHC risk area, and 20 maize samples from 20 farmers in the suburbs of Guilin (a low-risk area). AFB1 was detected in the majority of samples (85%), with higher concentrations (9–2496 µg/kg) in samples from the high-risk area. Among the samples, 76% exceeded the Chinese regulation of 20 µg/kg for AFB1 in maize intended for human consumption. The probable daily AF intake in the high-risk area was 3.68 µg/kg of body weight/day, which was three times the median toxic dose for rats. Results of this study confirmed that AFB1 plays an important role in the development of PHC in Guangxi.

Another comparative epidemiological study was conducted in China to identify the potential factors modulating AF exposure among three locations: Fusui County and Nanning City in Guangxi Province and Chengdu City in Sichuan Province [120]. These three locations had HCC rates of 92–97, 32–47, and 21 per 100,000 people, respectively. Residents were screened for AF-alb adducts and human papilloma virus (HPV) infection. Higher numbers of HPV-positive people (47%) were found among Fusui residents compared to Nanning (15%) and Chengdu (22%) residents. This suggests a co-effect of HPV infection and AFB1 exposure in the high risk of HCC in the Fusui region.

Taiwan is another province of China that has a high rate of dietary AF exposure and HCC prevalence. Wan et al. (1996) [121] published one of the earliest reports about AF-related HCC cases in Taiwan. They surveyed seven townships, including those with the highest HCC incidence. Detectable concentrations of the AF-alb adduct and urinary AF metabolites were highly correlated with HCC in 56 cases, and the authors concluded that AF exposure was enhancing the risk of HCC associated with HBV. In a similar study designed to elucidate the importance of AF exposure in the etiology of HCC, researchers conducted a community-based cohort epidemiological study in the Taiwan Penghu Islets, where the HCC mortality rate was highest [122]. In this study AF exposure was evaluated in inhabitants (6487) via regular follow-up. AFB1-albumin adducts were detected in 60% of HBsAg-positive HCC cases, and the authors concluded that a higher risk of developing HCC was attributable to both a heavy exposure to AFs and high HPV incidence.

Chu et al. (2017) [110] recently conducted a study to assess the effect of AFB1 exposure on cirrhosis and HCC in chronic HBV carriers in Taiwan. This case-controlled study was nested in a large community-based cohort that included seven townships. The researchers found that elevated serum AFB1-alb adduct levels were significantly associated with an increased risk of developing cirrhosis and cirrhotic HCC and with the risk of developing HCC in a dose-dependent manner in cirrhosis patients [110]. Additionally, Wu et al. (2009) [123] suggested that the combined effect of AF and HBV was additive rather than multiplicative based on a study conducted in Taiwan. 

Guangzhou is a city that is located in the southern part of China, where AF contamination is high due to warm and humid weather conditions that are favorable for the growth and proliferation of *A. flavus*. In a study aimed at describing the risks of dietary AF exposure, Zhang et al. (2020) [58] used the margin of exposure (MOE) and quantitative liver cancer risk approaches. The AFB1 content of 1854 food samples collected from 11 districts in Guangzhou was measured. In total, 9.9% of the test samples were positive for AFB1. Homemade peanut oil had the highest AFB1 concentration (38.74 ± 47.45 μg/kg). The MOE level of Guangzhou residents ranged from 100 to 1000. The risk of liver cancer was 0.0264 cases/year/100,000), and homemade peanut oil was the main contributor to dietary exposure to AFB1 for the residents. 

Liu and Wu (2010) [15] performed a quantitative AF-related HCC risk assessment based on China’s food consumption patterns, AF-contaminated food prevalence, HBV prevalence, and population size. They attributed about 5300–14,400 HCC cases in China each year to chronic AF exposure, whereas 1990–4430 HCC cases were attributed to synergetic effects of AFs and HPV. During the survey, the chronic HBV prevalence in China was among the highest in the world (8–10%) [124]. AF exposure of 17–37 ng/kg body weight/day was estimated based on total AF levels in staples such as cereals, as reported by different authors [49,50,55,118].

During the last 30 years, changes in China have led to a significant drop in AF-associated HCC prevalence and HCC mortality. This decrease might be due to socioeconomic changes and changes in the consumption pattern of maize, which dramatically decreased among Chinese families from 1980 to 2000 [125]. Additionally, after adoption of the national children’s HBV vaccination program (1980–1990), the prevalence of HCC has dropped significantly. Evidence for this decline comes from a recent cancer registration report showing about an 83% reduction of HCC mortality in Qidong, which was one of the regions with the highest prevalence [126]. The reduction of HCC incidence is also evident in results of a longitudinal (28 year) study that utilized follow-ups of etiological interventions among 1.1 million inhabitants of this area [126]. In this study there was a controlled neonatal HBV vaccination program (1980–1990) and economic reforms beginning in 1980 that were aimed at changing the consumption pattern from maize to rice and wheat. Compared with 1980–1983, the age-specific liver cancer incidence rates in 2005–2008 significantly decreased by 14-fold (ages 20–24), 9-fold (ages 25–29), and 4-fold (ages 30–34). The reduction among 20–24 year-olds might reflect the combined effects of reduced AF exposure and neonatal HBV vaccination, whereas the decreased incidence in the age groups of >25 years may be attributable mainly to a rapid reduction of AF exposure. 

## 5. AF Standards in China and Recent Updates 

AFs are the most regulated mycotoxins due to their toxicities and health risks, particularly carcinogenicity. Over 100 countries have defined maximum limits for AFs [30,31]. For cereals and nuts, most maximum limits range between 10 and 20 µg/kg, although the EU sets the lowest limit at 4 µg/kg [127]. Most countries set the maximum limits of AFM1 at either 0.05 µg/kg (EU) or 0.5 µg/kg. Currently, the Chinese government also has regulations on the maximum limits of AFs allowed in different foodstuffs. In maize, peanuts, peanut oil, nuts (walnuts, almonds), and dried fruit, the maximum limit of AFB1 is 20 μg/kg, whereas the limit in rice and oils (sesame, rapeseed, soybean, sunflower, flax, maize germ, bran, cottonseed) is 10 μg/kg. In milk and milk products (fresh raw milk, whole milk powder, evaporated milk, sweet condensed milk) and butter the AFM1 limit is 0.5 μg/kg, excluding liquid infant formula [128]. 

The establishment of mycotoxin legislation and regulations is dynamic both in terms of addressing newly identified toxins and maintaining rigorous limits. For instance, the EU has made the maximum tolerable limits more stringent over time [127,129]. It has been two decades since the Chinese government first imposed national food safety standards on maximum levels of mycotoxins in foods. Mycotoxin food standards were established in China in 2003 for the first time (GB 9676-2003) and considered toxins such as AFB1, AFM1, DON, and patulin, and they were renewed in 2005 (GB2761-2005). In 2011, GB2761-2011 added maximum limits of OTA and ZEN and considered baby foods and other special food groups [130]. In 2017, the Chinese maximum levels of mycotoxins in foods were updated, with a special emphasis on vulnerable groups of society (GB2761-2017). In this version, the maximum limit of AFB1 in infant formula and supplementary foods for infants, young children, and pregnant and lactating woman was set at 0.5 µg/kg. In 2019, GB 2761 lowered the maximum AFM1 limit from 0.5 to 0.2 µg/kg in liquid infant formula, which includes raw milk, pasteurized dairy, sterilized dairy, modified dairy, and fermented dairy. It also added a maximum limit for fumonisins (200 μg/kg) in cereal-based (maize or maize flour) auxiliary foods for infants [131]. Other common AFs (e.g., B2, G1, and G2) are being detected in different foods in China and are known to be carcinogenic, but they and total AFs are not yet regulated. 

## 6. Nature of the *Aspergillus* Species

Widely distributed fungi such as *A. flavus* and *A. parasiticus* are the primary members of the genus *Aspergillus* that account for preharvest and postharvest AF contamination of several agricultural commodities around the world. *A. flavus* strains are the most dominant fungi in different crops and agricultural products, both in the field and during storage. Hence, developing AF mitigation strategies has become a concern of many researchers worldwide. 

Understanding the nature and population structure of *Aspergillus* is crucial for developing AF mitigation strategies, and several studies focused on the distribution and nature of *Aspergillus* species have been conducted in different countries [14,37,38,132]. Studies have shown that *A. flavus* strains differ in their AF production capacity, as some of them produce different AFs and others may not produce any at all. Thus, researchers ultimately concluded that the A. flavus population consists of both AF producing and nonproducing strains, which are referred to as toxigenic and atoxigenic *A. flavus* strains, respectively [35,36,37,38].

Generally, the proportion of atoxigenic strains in the total *A. flavus* population varies among countries. For example, the proportion in Italy [43], Taiwan [42], and the USA [35,36] was <50%. In contrast, similar studies conducted in Nigeria [37,38], Argentina [41], Kenya [132], and Iran [4] reported that >50% of *A. flavus* isolates were atoxigenic.

Researchers have also explored the genetic makeup of strains that differ in AF production. Yu et al. (2005) reported clusters of biosynthesis genes involved in AF production for the first time [133]. In a later study of genomic function, researchers found that at least 30 genes and 20 enzymatic reactions are involved in AF biosynthesis [134]. These genes are clustered within a 75 kb DNA region in the chromosome 3 biosynthesis pathway, and most of them have been identified and sequenced [134,135,136] (Figure 1). Subsequent studies showed that those strains that do not produce AFs have either lost some genes or have specific mutations in the AF biosynthetic pathway. Such mutations were found to be caused by deletions of biosynthesis genes [45,48] or by insertions [48] and/or frameshifts [133] or single nucleotide polymorphisms [137] in genes involved in AF biosynthesis. These findings were supported by the presence of deletions, frameshift mutations, base pair substitutions, and termination point mutations in the AF synthesis pathways of two atoxigenic and domesticated varieties of *A. flavus* (*A. oryzae* and *A. sojae*) [138]. Researchers can use these findings to develop biocontrol agents from atoxigenic A. flavus strains that can be applied as a mitigation tool against toxin-producing strains either in the field or in storage.

## 7. Distribution and Genetic Characteristics of *Aspergillus* Species in China

The most important fungi responsible for AF contamination in China are the three members of *Aspergillus* section *Flavi*: *A. flavus, A. parasiticus*, and *A. nomius* [102]. Hence, understanding the distribution and mycotoxin (AFs and CPA) production capacities and the genetic makeup of these fungi is crucial for broader AF management and for designing preharvest and postharvest mitigation strategies. Interest in the distribution of *A. flavus* across different agro-ecologies, geographical locations, crops, and crop soils has also been increasing in China, due to the possibility of using isolates of atoxigenic *A. flavus* strains to reduce AF contamination. Thus, several reports describing the nature and distribution of *Aspergillus* section *Flavi* in China have been published, and they are summarized in Table 5. 

Gao et al. (2007) [39] identified *A. flavus* as the primary species responsible for AF contamination of maize in the northern parts of China. They found *Aspergillus* section *Flavi* isolates in 99% of 120 maize samples tested. Chen et al. (2019) [139] recently conducted a polyphasic analysis of the *Aspergillus* section *Flavi* population in maize (*n* = 195) collected from an AF-prevalent region (Guangxi Province). They reported that *A. flavus* strains were the dominant (98.5%) fungi, and among the 30 representative *A. flavus* isolates, the majority were found to produce both AFs and CPA. In contrast, only 6.7% were found to be non-AF and non-CPA producing strains. In another study, Wei et al. (2014) [46] isolated *A. flavus* strains from peanut fields in four provinces (Liaoning, Shandong, Hubei, and Guangdong). Of the 323 *A. flavus* strains detected, only 76 could not produce AFs. Furthermore, the incidence of atoxigenic strains decreased with increasing temperature and increased with increasing latitude.

Mamo et al. (2018) [40] isolated 724 *A. flavus* strains from maize, rice, and peanut kernels collected from all agricultural regions in four agro-ecological locations (19 provinces) in China. Only 229 (32%) of the *A. flavus* strains were atoxigenic, and 24 strains were atoxigenic and non-CPA [40]. In a survey of 600 soil samples collected from peanut fields in four agro-ecological zones (southeast coastal, Yangtze River, Yellow River, and the northeast) of China, Zhang et al. (2017) [47] isolated 344 *Aspergillus* isolates, of which 94.2% were *A. flavus* strains. The Yangtze River zone had the highest population density of *Aspergillus* sp. and the highest positive rate of AF production in isolated strains (1039.3 cfu/g, 80.7%). The lowest values occurred in the northeast zone (2.4 cfu/g, 6.6%). Lai et al. (2015) [141] studied the AF production potential of 127 *A. flavus* strains pre-isolated from rice collected from 12 provinces in China. After culturing each strain on rice at 28 °C for 21 days, 37% of them produced AFB1 and AFB2 up to a maximum of 124,101 and 10,329 µg/kg, respectively. 

In addition to studies of the distribution and chemotypes of *Aspergillus* isolates, several researchers assessed the genetic characteristics of *A. flavus* strains in China. Yu et al. (2019) [142] studied the genetic diversity of 88 toxigenic *A. flavus* strains isolated from 26 provinces in China. The phylogenetic tree classified the isolates into three populations: *A. flavus* I, *A. flavus* II, and *A. oryzae*. Most of the isolates in the first two populations produced AFs and CPA. Moreover, the study revealed that almost all of the *A. oryzae* isolates originated from northern parts of China [142]. Mamo et al. (2018)[40] reported that among the 229 atoxigenic strains identified from 724 *A. flavus* strains, 24 had deletions on three important AF biosynthesis genes (*aflR, fas-1*, and *aflJ*) and none of the 24 had PCR amplicons for five genes in the AF biosynthetic pathway. Moreover, 16 (67%) atoxigenic *A. flavus* strains were PCR-negative for 10 or more of the biosynthetic genes. Similarly, Wei et al. (2014) [46] found that 97% of the atoxigenic strains screened lacked at least one of the genes (*aflT, nor-1, aflR*, and *hypB*) involved in the AF biosynthetic pathway [46]. Furthermore, PCR amplification of all clustered genes revealed 25 deletion patterns, 22 of which were reported for the first time. Chang et al. (2009) [143] reported that 71% of the atoxigenic strains tested had deletions in two genes (*maoA* and *dmaT*), which are members of the CPA biosynthetic cluster. A similar PCR analysis showed that 12 AF biosynthesis genes (*aflT, pksA, nor-1, fas-2, fas-1, aflR, aflJ, adhA, estA, norA, ver-1*, and *verA*) were deleted in a potential *A. flavus* biocontrol strain (GD-3) that was isolated from a peanut field in Guangdong Province [44].

Atoxigenic *A. flavus* strain AF051 was isolated from a peanut field in Huaian, Jiangsu Province, China in 2006 and has shown potential for AF biocontrol [48]. Molecular characterization of this strain confirmed that it does not generate fragments for genes *norB, cypA, aflT, pksA, aflR*, and *norA* in the AF biosynthesis pathway. This strain also has an 89.59 kb deletion in the AF gene cluster, which is replaced by a 3.83 kb insert. The dissimilarity between the insert fragment and any gene present in known toxigenic strains might prevent reversion when it is used as a biocontrol agent. 

Yan et al. (2018) [140] analyzed genetic diversity of *A. flavus* isolates from peanut kernels. They reported that 38.0% of isolates lacked the genes *nor-1, ver-1, aflR*, and *omtA* from the AF biosynthesis pathway. They also identified 49 single nucleotide polymorphisms in a 1254 bp fragment of the *omtA* gene, which illustrated genetic variation of the *omtA* gene among different *A. flavus* isolates. 

In a recent study, Chen et al. (2020) [73] described the distribution of *Aspergillus* species in herbal medicines in China. They found that isolates of *Aspergillus* and species of *Penicillium, Rhizopus*, and *Trichoderma* were the predominant fungi present in the samples. Among the six *A. flavus* strains isolated from medicinal herbs, only one found in Amomi fructus was able to produce AFB1 and AFB2, and only two amplified the genes *aflR, omt-1, nor-1*, and *ver-1* during PCR analysis. Additionally, several studies reported diversified deletion patterns (1.5 kb or 1.0 kb) in the *norB–cypA* intergenic region of the AF biosynthetic gene cluster of different *A. flavus* strains in China [40,45,46,140].

## 8. Atoxigenic *A. flavus* as AF Biocontrol Agents

Recent developments in the field of mycotoxin prevention have led to renewed interest in seeking effective, feasible, and environmentally friendly control strategies. Biological methods, which basically utilize naturally occurring microorganisms or their enzymes or extracts, have been confirmed to have fewer food safety problems and environmental impacts compared to chemical methods. Biological control methods are also promising because they are very specific for specific toxins. Over the last three decades, numerous investigations have been conducted to explore potential biological control agents (e.g., fungi, bacteria, enzymes, and proteins) that can reduce mycotoxin levels either by inhibiting fungal growth and/or proliferation and subsequent mycotoxin production, or by degrading (transforming) them into harmless metabolic products.

The application of atoxigenic *A. flavus* is now becoming a widely applicable biocontrol mechanism. The effect is achieved by applying naturally occurring competitive native atoxigenic strains of *A. flavus* to the soil [144]. Atoxigenic *A. flavus* strains interfere with the proliferation of indigenous toxigenic strains [13,145,146,147,148]. Atoxigenic strains that are inoculated in the soil have been shown to have a carry-over effect that may inhibit peanut contamination during storage [149], which makes the method more acceptable because it may reduce AF contamination during both preharvest and postharvest. Several countries (USA, Italy, Argentina, Nigeria, Australia, and Thailand) are either developing or already using this native biocontrol agent widely [13,145,146,147,148]. In doing so, they have achieved significant levels of AF reduction (43–98%). Several atoxigenic strains of *A. flavus* have been patented, registered, and commercialized. In the USA between 2004 and 2008, the atoxigenic *A. flavus* strains NRRL 21,882 (active component of Afla-guard^®^) and AF36 (NRRL 18543) were registered and used [150] widely. Strain K49 (NRRL 30797) was also patented by the USDA [151]. 

In order to develop sustainable AF biocontrol, the population dynamics and genetic stability of *A. flavus* populations in the field must be carefully examined. Due to the possibility of recombination with toxigenic strains, atoxigenic *A. flavus* strains could develop the ability to produce AFs [152]. Therefore, it is critical to assess the frequency of such events in agricultural environments where atoxigenic biocontrol *A. flavus* has been introduced [153]. Analysis of vegetative compatibility groups (VCGs) is critical, as VCGs are a strong barrier to sexual recombination [154]. Atoxigenic biocontrol isolates selected for use should belong to VCGs that contain only atoxigenic strains and have wide distributions [155], as different VCGs are clonal lineages that differ in many characteristics, including AF-producing ability [156]. 

### AF Biocontrol Developments in China

In China, several strategies have been developed to manage and control AFs. These methods, which aim to either prevent or control AFs, include breeding for resistance, field management practices, proper storage, chemo-prevention, adsorption and detoxification, and dietary changes. Post-harvesting techniques, such as sorting, cleaning, fast and proper drying, insect control, spraying, and smoking with synthetic pesticides or botanicals have been used as storage protectants [102]. Physical absorption [157,158], thermal inactivation [159,160], irradiation [161], and chemical treatments [162,163] have been used to detoxify or inactivate mycotoxins. However, these mitigation strategies have drawbacks related to safety issues, impacts on nutritional value, limited efficacy, cost implications, and environmental impacts. 

At the global level, the application of *A. flavus* as an AF biocontrol agent became a promising AF mitigation strategy in the mid-2000s. During the last decade, a few studies in China have reported the efficacies of native *A. flavus* strains as AF biocontrol agents either in the laboratory or under field conditions [40,44]. Yan et al. (2021) [164] recently tested the efficacy of atoxigenic *A. flavus* strains against high AF-producing strains co-inoculated at equal amounts in the soil in China, and they achieved a significant AF reduction (84.96–99.33%). However, their effect has not been tested under field conditions against the naturally occurring multiple fungal strains. Moreover, the carry-over effects of field-applied atoxigenic *A. flavus* on stored peanuts have yet to be reported. Preliminary experiments showed that atoxigenic strain AF051 was highly competitive against toxigenic strains in peanut fields, and the incidence of this strain (12%) was higher than that of other atoxigenic strains in the population of atoxigenic strains collected from peanut fields [48].

Mamo et al. (2018) [40] reported large AF reductions of 82.8% (SXN, strain from Shannxi), and 87.2% (JS4, strain from Jiangsu) achieved by atoxigenic strains applied in a commercial peanut field in Guangdong, and their displacement abilities and AF reductions were promising. Laboratory-based competitive experiments also indicated significant AF reduction (Mamo, 2018 unpublished data). Zhou et al. (2015) [44] described AF reductions ranging from 33% to 99% in a laboratory-based competitive experiment in which the atoxigenic strain GD-3 was co-inoculated into the experimental system to act against the toxigenic strain, and the reduction was correlated with the competitor ratio. These results demonstrated that GD-3 was successful at reducing AF contamination, thus it shows promise as a potential biocontrol agent for local farmers.

## 9. Conclusions and Recommendations

Historically, mycotoxins, especially AFs, were the most prevalent toxins in China. The incidence of HCC cases due to AF exposure was also among the highest in China compared to other countries. Some regions of China are particularly prone to AF contamination due to the climate and geographical location. However, strong Chinese AF regulations, control measures, such as good agricultural and manufacturing practices that include preventive strategies from preharvest to postharvest, policy implementations, and national HBV vaccinations are yielding encouraging results in terms of reduced AF prevalence and exposure risks. Nonetheless, as revealed in some reports reviewed herein, the prevalence of AFs in some agricultural products is still relatively high compared to the national maximum limits. Thus, there is still a need for extensive surveys of AF contamination in food and foodstuffs as well as human biomarker monitoring in order to improve risk management. 

Feasible mechanisms to reduce toxic AF levels throughout the food chain, from farm to storage and processing, are needed. The application of native atoxigenic *A. flavus* as a biocontrol mechanism is currently being used in different countries, as it is easy to use and environmentally friendly. Soil treatment with atoxigenic strains offers the extra advantage in the carry-over effect of reducing AF contamination that occurs during storage. According to several reports from China, several indigenous atoxigenic *A. flavus* can be used as biocontrol agents. However, little is known about the potential of these native strains at the experimental level or under field conditions. The few existing studies suggest that these native strains show a promising competitive ability to displace toxigenic strains and reduce AF levels [41,47,142]. Thus, there is a need for more studies to characterize and search for potential AF biocontrol strains of *A. flavus* and to develop formulations and application techniques for these biocontrol strains in the field. Extensive field trials must be conducted across China in areas where potential AF contamination risks are high. Finally, VCG tests should be conducted to ensure genetic stability of atoxigenic strains and to test for their recurrent efficiency. 

## Figures and Tables

**Figure 1 toxins-13-00678-f001:**
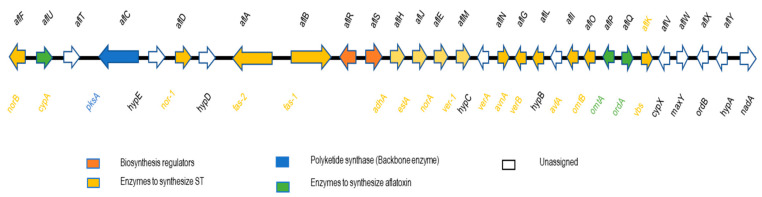
Aflatoxin biosynthesis genes.

**Table 1 toxins-13-00678-t001:** Aflatoxin contamination in cereal and cereal-based foods reported in China.

Province	Crops	Origin	Period	Test	Detection Limit (µg/kg)	Mycotoxin	Total Samples	Incidences (%)	Range of Positive Samples/Maximum Value (µg/kg)	Mean ± SEM of Positive Samples/Mean (μg/kg)	Level of Contamination above Chinese Regulatory Limit (%) (µg/kg)	References
Liaoning	Maize	Farmer stores	2003	HPLC		AFs	73	97	-	0.99	All < 20	[52]
	Whole grain rice						16	100	-	3.87		
	Brown rice						37	97	-	0.88		
Heilongjiang	Rice	Farmer stores, granaries, and markets	2009–2011	DLLME HPLC		AFs	62	69	0.033–0.17	0.062 ± 0.042	All < 20	[53]
Liaoning							30	96.7	ND	ND		
Jilin							59	40	0.030–0.98	0.12 ± 0.25		
Guangdong							138	53	0.19–4.1	0.44 ± 0.90		
Guangxi							67	81	0.032–21	1.3 ± 3.7		
Hainan							14	93	0.032–0.71	10.23 ± 0.32		
Heilongjiang					LOQ = 0.009	AFB1	62	69.3	0.033–0.14	0.058 ± 0.034		
Liaoning							30	97	<LOQ	<LOQ		
Jilin							59	39	0.030–0.90	0.11 ± 0.23		
Guangdong							138	73	0.030–3.7	0.41 ± 0.81		
Guangxi							67	53	0.032–20	1.2 ± 3.4		
Hainan							14	93	0.032–0.66	0.21 ± 0.30		
Heilongjiang					LOQ = 0.006	AFB2	62	14.5	0.022	0.022		
Liaoning							30	6.6	<LOQ	<LOQ		
Jilin							59	6.5	0.086	0.086		
Guangdong							138	13	0.020–0.47	0.11 ± 0.15		
Guangxi							67	37.3	0.029–1.6	0.19 ± 0.36		
Hainan							14	14.3	0.051	0.051		
Shandong Province (Huantai County)	Maize	Individual households	2010	ELISA	0.1	AFB1	31	100	0.4–2.2			[54]
	Rice						9	100	0.1–1.2			
	Wheat flour						9	100	0.3–0.9			
Jiangsu Province (Huaian City)	Maize						43	100	1.2–136.8			
	Rice						10	100	0.2–0.7			
	Wheat flour						7	100	0.1–0.3			
Guangxi Zhuang Autonomous (Fusui County)	Maize						34	100	1.0–50.0			
	Rice						10	100	0.3–1.4			
	Wheat flour						-	-	-			
Eight regions (Chongqing, Fujian, Guangdong, Guangxi, Hubei, Jiangsu, Shanghai, Zhejiang)	Maize	Local food markets	2007	HPLC	0.012 ;B1 0.008; B2; 0.036;G1	AFs	74	52	0.02–1098.36			[55]
						AFB1	74	46	0.14–970.32		23.91 above 20	
						AFB2	74	41	0.02–128.04			
						AFG1	74	9	0.36–4.76			
	Rice					AFs	84	23	0.15–3.88			
						AFB1	84	16	0.15–3.22			
						AFB2	84	3	0.06–0.24			
						AFG1	84	7	0.36–1.59			
Yangtze Delta region (Hangzhou, Ningbo, Shanghai, Suzhou and Wuxi cities)	Rice, wheat, maize, oats, soya bean	Supermarkets and wholesale markets	2010	IAC-fluorometer	1	AFs	76	14.5	1.1–35.0	6.9	4.0 beyond Chinese (20) and 6.6 beyond EU(4)	[51]
					1	AFB_1_	76	14.5	1.0–32.2	6.6	4 beyond Chinese limit (20) and 9.2 beyond EU limit (2)	
Hangzhou	Cereal based infant food	Supermarkets	2012	UPLC-MS/MS	0.001	AFB1	30	6.6	0.016–0.024			[56]
					0.001	AFB2		0	ND			
					0.002	AFG1		0	ND			
					0.006	AFG2		0	ND			
					0.008	AFM1		0	ND			
					0.004	AFM2		0	ND			
Chongzuo County and Guilin suburbs, Guangxi autonomous region	Maize	Individual households	1998	HPLC	1	AFB1	40	45	9–2496	460 ± 732	76% AFB1positive samples above Chinese limit (20)	[50]
					2.5	AFB2		35	11–320	82 ± 102		
					10	AFG1		22.5	12–21	15 ± 3		
					10	AFG2		0	ND	ND		
Taiwan	Coffee, red yeast rice and maize	Local stores	2013	ELISA	1–2	AFB1	36	55.5	1.7–234.0		30% are beyond the Taiwan limit (15)	[57]
Eleven districts of Guangzhou	Rice and rice products	Household supply retail shops	2015–2017	HPLC	0.1	AFB1	490	1.42	0.28–1.00	0.13 ± 0.001		[58]
	Wheat and wheat products						436	1.4	0.28–1.46	0.13 ± 0.001		
							339	0.9	1.50–6.30	0.17 ± 0.001		
	Maize and maize products											
Guangxi ; Zhuqing Village, Fusui,	Maize	Households	1999	ELISA	-	AFB1	30	76.7	0.4–128.1	23.7 ± 6.6	30% beyond (20)	[49]
	Rice						30	23.3	0.3–2.0	1.1 ± 0.3		
Shigatze Prefecture of Tibet Autonomous Region	Barley	Farms	1998	CD-ELISA		AFs	25	4 0.0	-	0.04		[59]

**Table 2 toxins-13-00678-t002:** Aflatoxin contamination in peanut, pine nut, walnut, other oil seeds and oil reported in China.

Province		Crops	Origin of Sample	Period	Analytical Method	Detection Limit (µg/kg)	Mycotoxin	Total Samples	Incidences (%)	Range of Positive Samples/Maximum Value (µg/kg)	Mean ± SEM of Positive Samples/Mean (μg/kg)	Level Contamination above Chinese Regulatory Limit (%)	References
Twelve provinces, including Liaoning, Shandong, Henan, Hebei, Jiangsu, Anhui, Jiangxi, Hubei, Hunan, Guangdong, Guangxi, and Fujian		Peanut with pod	From farm	2011/2012	HPLC	3:1 for LOD	AFB1	1040	25	0.01–720	2.13	1% beyond Chinese regulation (20) and 3.7% above EU regulation (2)	[61]
Liaoning		Peanut	From farm and storage	2015	HPLC	0.2 for AFB1; 0.05 for AFB2; 0.2 for AFG1; 0.05 for AFG2	AFB1	408	3.19	0.15–116.64	0.43 ± 6.23	-	[32]
							AFB2		3.68	0.05–27.36	0.11 ± 1.50		
							AFG1		0.25	3.61	0.01 ± 0.18		
							AFG2		0.74	0.27–1.15	0.00 ± 0.06		
							Total AF		4.90	0.05–144.00	0.55 ± 7.80		
Henan							AFB1	1190	19.00	0.06–483.00	7.57 ± 41.12		
							AFB2		11.68	0.01–61.50	0.82 ± 4.67		
							AFG1		1.18	0.33–460.00	0.81 ± 15.23		
							AFG2		4.03	0.05–104.00	0.23 ± 3.33		
							Total AF		19.00	0.06–1023.2	9.43 ± 54.98		
Sichuan							AFB1	455	15.60	15.56 ± 86.73	15.56 ± 86.73		
							AFB2		13.19	2.34 ± 13.40	2.34 ± 13.40		
							AFG1		0.22	0.07 ± 1.57	0.07 ± 1.57		
							AFG2		5.27	0.22 ± 1.82	0.22 ± 1.82		
							Total AF		15.60	18.19 ± 100.38	18.19 ± 100.38		
Guangdong							AFB1	441	11.56	0.22–341.41	4.73 ± 29.84		
							AFB2		11.79	0.05–30.38	0.51 ± 2.96		
							AFG1		0.91	0.50–11.50	0.04 ± 0.57		
							AFG2		3.17	0.21–5.74	0.06 ± 0.41		
							Total AF		14.29	0.06–373.69	5.34 ± 32.90		
Eight regions (Chongqing, Fujian, Guangdong, Guangxi, Hubei, Jiangsu, Shanghai, Zhejiang)		Peanut	Local food markets	2007	HPLC	-	Total AF	65	15	0.03–28.39		Average 27.44	[55]
		Walnut						48	31	0.02–1.20			
		Pine Nut						12	2	0.19–0.25			
		Peanut					AFB1	65	9	0.15–22.39			
		Walnut						48	21	0.14–0.32			
		Pine Nut						12	2	0.19–0.23			
		Peanut					AFB2	65	5	0.03–6.00			
		Walnut						48	12	0.02–0.70			
		Pine Nut						12	1	0.02			
		Peanut					AFG1	65	4	0.42–11.73			
		Walnut						48	8	0.36–0.83			
		Pine Nut						12	0	-			
Shandong Province (Huantai County), Jiangsu Province (Huaian City), and Guangxi Zhuang Autonomous (Fusui County)		Plant oil	Individual households	2010/2011	ELISA	0.1	AFB1	39	100	0.5–114.4		Median level is 52.3 beyond the Chinese standard 10	[54]
		Peanut						17	100	0.1–0.7			
Hebei Province	Shijiazhuang	Edible oil (peanut, blended, soybean, maize, sunflower, fish oil)	Local markets	2011	LC–MS/MS		AFB1	40	32.5	0.14–2.72			[62]
							AFB2		12.5	0.15–0.36			
							AFG1		7.5	0.01–0.02			
	Baoding						AFB1	18	22.2	0.16–1.88			
							AFB2		5.56	0–0.18			
	Tangshan						AFG1		0	-			
							AFB1	18	27.8	0.15–0.45			
							AFB2		0	-			
							AFG1		0	-			
Beijing, Shanghai, Changchun, Chengdu, Shijiazhuang, and Zhengzhou		Peanut butter	Retail markets	2007	LC	1	AFT	50	82	0.77–70.64	8.51	39% for total AFs set by EU (4) 37% AFB1 set by EU (2) and 2% AFB1 exceed the Chinese regulations (20);	[63]
						0.15	AFB1			0.39–68.51	6.12 6		
							AFB2			0–5.52	0.67		
							AFG1			0–21.22	2		
							AFG2			0–6.36	0.4		
		Sesame paste				1	AFT	50	37	0.54–56.89	6.75	24% beyond the limits total AFs of EU (4) 19% and 32% of sesame AFB1 exceed Chinese (5) and European Union (EU) (2)	
						0.15	AFB1			0.39–20.45	4.31		
							AFB2			0–4.92	0.63		
							AFG1			0–26.28	1.44		
							AFG2			0–5.75	0.37		
Eleven districts of Guangzhou		Nuts	Household supply retail shops	2015–2017	HPLC	0.1	AFB1	96	3.1	0.62–1.37	0.14 ± 0.001		[58]
		Vegetable oil						365	38.9	0.26–283.0	6.32 ± 25.99		
		Commercial vegetable oil						269	25	0.35–7.30	0.67 ± 1.81		
		Home-made peanut oil						96	75.5	0.26–283.0	38.74 ± 47.45	The mean Is 7 times larger that the Chinese maximum limit (5)	
21 provinces, autonomous regions and municipalities		Nuts	Local markets and supermarkets	2018	UPLC	LOD; 0.05–1.00; LOQ; 0.10–5.00	AFB1	133	3.8	1.3–40.7	9.3 ± 0.28		[64]
							AFB2		15	0.2–1.2	1.9 ± 0.02		
							AFG1		ND	ND	ND		
							AFG2		2.3	1.1–1.6	1.3 ± 0.02		
Guangxi; Zhuqing Village, Fusui,		Peanut	Households	2013	ELISA	-	AFB1	30	66.7	0.1–52.5	7.8 ± 3.2		[49]
Yangtze Delta region (Hangzhou, Ningbo, Shanghai, Suzhou and Wuxi cities)		Peanut, soya bean, and oil.	Supermarkets and wholesale markets		IAC-fluorometer	1	AFs	76	14.5	1.1–35.0	6.9	4.0	[51]
							AFB_1_	76		1.0–32.2	6.6		

**Table 3 toxins-13-00678-t003:** Aflatoxin contamination in Chinese herbal medicine, spices, tea, fruits and vegetables reported in China.

Province	Product	Origin of the Sample	Study Year	Analytical Method	Mycotoxin	Detection Limit (µg/kg)	Total Samples (*n*)	Incidences (%)	Range of Positive Samples/Maximum Value (µg/kg)	Mean ± SEM of Positive Samples/Mean (μg/kg)	Level Contamination above Chinese Regulatory Limit (%)	References
Eleven districts of Guangzhou	Tea	Household supply retail shops	2019/2020	HPLC	AFB1	0.1	128	17.9	0.25~4.0	0.36 ± 0.62		[58]
Anhui, Fujian, Gansu, Guangdong, Guizhou, Hubei, Shanxi, Xinjiang, Yunnan, Zhejiang	Traditional Chinese medicines (TCM)	Herbal market	2019/2020	HPLC	AFB1	0.012–1.3	48	70.8	0.12–3.05		All < 5	[73]
					AFB2				0.43–0.5		AFB1 limit 2–10, AF’s ; 4–20 (Chinese AFB1 ≤ 5 ; AFs ≤ 10)	
					AFG1				ND–0.85			
					AFG2				0.87–2.11			
China	TCM	Regulated enterprises	2011	UHPLC/MS/MS	AF’s	LOD; 0.01–1.56	60	40	0.2–19.5			[75,76]
					AFB1				1.2–9.8			
					AFB2				0.2–7.1			
					AFG1				0.6–2.5			
					AFG2				0.2–4.8			
Chongqing China	TCM	Local markets and drug stores	2015	UPLC-MS/MS	AF’s	LOD; 0.008–0.022	22	63	0.2–7.5		18.2 exceeded the maximum limit set by EU (4)	[77]
					AFB1				0.2–4.8			
					AFB2				0.1–2.3			
					AFG1				0.1–0.8			
					AFG2				0.1–0.2			
Zhejiang	TCM	Regulated enterprises	2009/2010	(UHPLC–MS/MS	AFB1	LOD; 0.01–1.56	30	68.8	-	1.40		[78]
					AFB2			50.0		1.27		
					AFG1			43.8		0.50		
					AFG2			43.8		0.94		
					AFM1			6.6		0.7		
Beijing	Ginger	Local markets	2013/2014	UHPLC-FLR	AFB1	0.005–0.2	30	5/30	0.3–1.38	0.073		[79]
					AFB2		30	ND	-			
					AFG1		30	ND	-			
					AFG2		30	ND	-			
Hebei province and Guangxi provinces	Chinese yam, American ginseng, Ginseng, Notoginseng, Astragalus, Polygala, Bupleurum, Liquorice	Markets	2013	UPLC-MS/MS	AFB1	LOD ≤ 0.05 and LOQ ≤ 0.1	48	35.4	ND-13.3		14.58 exceed 5	[80]
					AFB2			2	ND-8.2			
					AFs			37.5	ND-21.5		8.33 exceed 10	
Shanghai	Pistachios	Markets	2014–2015	LC-MS/MS	AFB1	0.03	25	4	ND-0.8	0.8	-	[81]
					AFB2	0.2		0	ND	ND		
					AFG1	0.2		0	ND	ND		
					AFG2	0.3		0	ND	ND		
	Dried longans				AFB1	0.1	28	0	ND	ND		
					AFB2	0.1		3.6	ND-0.2	0.2		
					AFG1	0.2		0	ND	ND		
					AFG2	0.3		0	ND	ND		
	Raisins				AFB1	0.1	32	0	ND	ND		
					AFB2	0.3		0	ND	ND		
					AFG1	0.3		0	ND	ND		
					AFG2	0.3		0	ND	ND		
	Dried dates				AFB1	0.1	40	0	ND	ND		
					AFB2	0.1		0	ND	ND		
					AFG1	0.3		0	ND	ND		
					AFG2	0.3		0	ND	ND		
21 provinces, autonomous regions and municipalities	Dried jujube	Local markets and supermarkets	2018	UPLC-MS/MS	AFB1	LOD; 0.05–1.00 and LOQ; 0.10–5.00	35	0	ND	ND		[64]
					AFB2			0	ND	ND		
					AFG1			8.6	0.2–0.6	0.4 ± 0.03		
					AFG2			2.9	0.4	0.4 ± 0.06		
	Raisins				AFB1		30	0	ND	ND		
					AFB2			0	ND	ND		
					AFG1			0	ND	ND		
					AFG2			20	0.5–1.4	0.9 ± 0.02		
	Dried figs				AFB1		20	15	1.8–384.1	129.5 ± 0.68		
					AFB2			5	2.5	2.5 ± 0.21		
					AFG1			15	0.4–17.8	5.9 ± 0.33		
					AFG2			15	0.6–1.2	0.9 ± 0.05		
	Dried longans				AFB1		15	ND	ND	ND		
					AFB2			6.7	0.7	0.7 ± 0.01		
					AFG1			ND	ND	ND		
					AFG2			40	0.1–2.9	1.5 ± 0.07		

**Table 4 toxins-13-00678-t004:** Aflatoxin contamination of animal feeds, and dairy products reported in China.

Province	Product	Year	Origin of the Sample	Analytical Method	Mycotoxin	Detection Limit (µg/kg)	Total Samples (*n*)	Incidences (%)	Range of Positive Samples/Maximum Value (µg/kg)	Mean ± SEM of Positive Samples/Mean (μg/kg)	Level Contamination above Chinese Regulatory Limit (%) (µg/kg or L)	References
Ten provinces (Heilongjiang, Inner Mongolia Beijing, Tianjin Ningxia, Hebei, Shanxi, Shandong, North Shanghai, Guangdong, South)	Dairy cow feeds	2013	Dairy farms	HPLC	AFB1		200	42	0.05–3.53	0.31	<10	[91]
					AFB2		200	36	0.03–0.84	0.14	-	
					AFB1 + AFB2		200	24.5	0.05–3.53	0.34	-	
	Milk		Dairy farms	ELISA	AFM1	0.005	200	32.5%	5.2–59.6 ng/L	0.0153	<0.5	
Beijing	Feed and feedstuffs	2012	Animal farms	HPLC	AFB1	-	22	50	59	6.0	<Chinese limit	[92]
					AFB2			9.1	12	0.6		
					AFG1			4.5	0.5	0.0		
					AFG2			9.1	0.5	0.0		
31 provinces	Yoghurt	2013	Retail store and supermarkets	ELISA	AFM1	0.05 μg/kg	178	4.49	-	27.10	<0.5	[93]
	Milk					0.005 μg/kg	233	48.07	-	21.49	<0.5	
Tangshan region of China	Milk	2012–2014	Milk stations	HPLC-MS/MS	AFM1		530	52.8%	10–200 ng/L	73.0 ng/L	<0.5	[94]
China	Feed		Company and livestock farms	Eu-Nano-TRFIA	Total AFs	0.16 μg/kg	397	78.3%	0.50–145.30 μg/kg			[95]
Northern China	Raw milk	2019/2020	Shops, distributors, farms	ELISA	AFM1	-	84		10–430 ng/L	110 ng/kg	34.5% exceeds EU limits	[96]
	Commercial milk				AFM1		69					
	Total mixed rations (TMR)			HPLC	AFB1	0.03 μg/kg	22		30–370 ng/L	4.16 μg/kg	31.8% exceeds EU limits	
Central China	Feed	2016/2017		HPLC	AFB1	0.03 μg/kg	174	35.1%			2.3% (30)	[97]
	UHT milk			ELISA	AFM1	0.005 μg/kg	111	73.6%	-	100.0 ng/L	All below 0.5	
	Pasteurized milk			ELISA	AFM1		131		-			
China (Beijing and Shanghai)	UHT milk	2010	super- markets	ELISA	AFM1	-	153	54.9%	0.006–0.160 mg/L	-	All below 0.5	[98]
	Pasteurized milk					-	26	96.2%	0.023–0.154 mg/L	-	20.3% of UHT milk samples and 65.4% of pasteurized milk samples exceed the EU limit	

**Table 5 toxins-13-00678-t005:** Distribution of *Aspergillus flavus* reported by different authors in China.

Location		Product	Number of Samples	Source	Sampling Season	Incidences of Fungi, *or Aspergillus* spp. or *Aspergillus* Section *Flavi*	Incidences of *A. flavus* Species		Toxin Production of *A. flavus* Strains	Nature Biosynthetic Genes	Morphological Nature	Reference
Liaoning Province (Northeast)		Maize	120	Household stored (1–3 years)	2003	55.8% (*Aspergillus* section *Flavi*)	98.5%		-	-	64%-L 36%-S	[39]
Guangxi province		Maize	89		2016/2017	195 (*Aspergillus* section *Flavi*)	98.5%		86.6% (30) were aflatoxin and CPA positive	-	Fluorescence and pink color observed in carbon added PDA	[139]
19 provinces, 1 autonomous region and 1 municipality		Peanut, maize, rice	-	-	2013/2014	724 *A.flavus* species isolated	>95%		32% (229) atoxigenic	10.4% atoxigenic strains found to have lost *aflR, fas-1* and *aflJ* genes	51% S-type (229) 34% L- type (229)15% NS (229)	[40]
14 provinces		Peanut pod	1106		2013	265 *Aspergillus* spp.	262 (98.9%)		18.8% *A.flavus* atoxigenic	38.0% atoxigenic strains lost *nor-1, ver-1, aflR, omtA* genes	-	[140]
					2015	257 *Aspergillus* spp.	254 (98.8%)					
12 provinces		Rice	-	-	-	-	127 *A.flavus*		47(37%) toxigenic	-	-	[141]
Provinces	Liaoning	Peanut-cropped soils	-	Field	2013	343 fungi isolated	9	323	76 Atoxigenic	[46]	97% of atoxigenic strains lost one of the *aflT, nor-1, aflR, hypB* genes	
	Shandong						73					
	Hubei						125					
	Guangdong						116					
Different provinces of China		Peanut cropped soil	-	Field	-	-	56 *A.flavus*		35 atoxigenic	11 *A. flavus* isolates had 5 deletion patterns for 12 genes	21 atoxigenic strains were either L- or S-type	[45]

## Data Availability

No new data shown in this paper.

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
