# Peer review of "Distribution of Aspergillus Fungi and Recent Aflatoxin Reports, Health Risks, and Advances in Developments of Biological Mitigation Strategies in China"

_toxins, 2021, doi:10.3390/toxins13100678_

Round 1

Reviewer 1 Report

Review of the manuscript "Distribution of Aspergillus fungi and recent aflatoxin reports, health risks, and advances in developments of biological mitigation strategies in China".
The assumptions of the authors of the manuscript were: "We present an update of information about AF contamination in different food commodities, AF distribution, molecular characteristics of the main AF-producing molds, AF-related health risks, and recent developments in AF biocontrol as mitigation strategies in China. We also discuss future perspectives. "
The manuscript contains a lot of interesting data collected from various scientific studies. However, the manuscript editing form is not adequate to the assumptions. The manuscript also contains numerous errors.
I will quote a few of them.
- no explanation of abbreviations when used for the first time. (e.g. IARC)
- alternately describing contamination with toxic fungi and aflatoxins
- errors in the values ​​describing the maximum aflatoxin content in spices and herbs in the EU (the maximum limit is 5.0 and 10.0 μg / kg) - line 243
- "animal feeds should be free of fungal and AF contaminations" -line 251 - what legal acts describe the requirements for the absence of fungi?
- the number of samples with determined values ​​of aflatoxin B1 in the EU imported from China is described annually in the RASSF report (Rapid Alert System for Food and Feed) - the authors did not use any data from this report.
-chapter "AF detection in China" -line 269 - has no value. It was necessary to provide at least the levels of the detection limit for individual techniques, or the possibility of multi-component analysis.
- the "AF biocontrol developments in China" section contains selected data describing "atoxigenic strains". However, this is not an implementation of the assumption of "recent developments in AF biocontrol as mitigation strategies". That's just what's missing from the manuscript.
In conclusion, although the manuscript contains a large amount of data, it needs to be re-edited.

Author Response

Thank you! We found your comments extremely helpful and have revised accordingly. Our responses are given in a point -by-point manner below. Changes to the manuscript are shown by a green highlight

 Responses to reviewers’ comments

Reviewers 1

Review of the manuscript "Distribution of Aspergillus fungi and recent aflatoxin reports, health risks, and advances in developments of biological mitigation strategies in China".
The assumptions of the authors of the manuscript were: "We present an update of information about AF contamination in different food commodities, AF distribution, molecular characteristics of the main AF-producing molds, AF-related health risks, and recent developments in AF biocontrol as mitigation strategies in China. We also discuss future perspectives. "
The manuscript contains a lot of interesting data collected from various scientific studies. However, the manuscript editing form is not adequate to the assumptions. The manuscript also contains numerous errors.

Point -by -point responses to comments by reviewers -1

Comment- 1. - no explanation of abbreviations when used for the first time. (e.g. IARC)

Responses: As to your valuable comments the abbreviations IARC has defined in its full name “International Agency for Research on Cancer” in its first appearance in page 2-line number 47.
Comments-2. - alternately describing contamination with toxic fungi and aflatoxins

Responses: Your valuable comments recognized and corrected throughout the text
Comment-3. - errors in the values ​​describing the maximum aflatoxin content in spices and herbs in the EU (the maximum limit is 5.0 and 10.0 μg / kg) - line 243

Response: EU countries have established standards, which sets the maximum limits of AFB1 (2 μg/kg) and total AFs (4 μg/kg) as stated in the reference number [74] (http://dx.doi.org/10.1016/j.fgb.2014.02.005 ) table 1 and page 4. 
Comment-4. - "animal feeds should be free of fungal and AF contaminations" -line 251 - what legal acts describe the requirements for the absence of fungi?

Response: According to your important comment. Legal acts restricted the fungal (mold) load might be different across different countries as well as across different feed types. However, in general the recommended ranges of fungal load on specific feed type like silage has been incorporated. Similarly, utilization of the term “contamination” interchangeably with fungi and aflatoxin has been corrected in line 250-253. 
Comment-5. - the number of samples with determined values ​​of aflatoxin B1 in the EU imported from China is described annually in the RASSF report (Rapid Alert System for Food and Feed) - the authors did not use any data from this report.

Response: Thanks for your insightful comment. According to RASFF report the main food item from China resulting several notifications due to excessive aflatoxin is peanut and peanut product. Accordingly, I have summarized a recently posted aflatoxin related notifications (2020-2020) from RASFF portal.

Change: The change is shown in line 218-224 in green highlight.
Comment -6. -chapter "AF detection in China" -line 269 - has no value. It was necessary to provide at least the levels of the detection limit for individual techniques, or the possibility of multi-component analysis. Response: Thanks for your nice comment. The section is aimed to show the common types of aflatoxin detection methods currently used in China. As of your comment the detection limits of each technique should be of our main focus. However, I omitted it as the detection limits of each method reviewed in this paper are indicated in the tables (1,2,3, and 4).
Comment-7. - the "AF biocontrol developments in China" section contains selected data describing "atoxigenic strains". However, this is not an implementation of the assumption of "recent developments in AF biocontrol as mitigation strategies". That's just what's missing from the manuscript.

Response:  Thanks again for your nice comment, the paper is basically aimed to show the current prevalence of aflatoxins, the distribution of main aflatoxin producing fungi (Aspergillus flavus) and the developments of AF biocontrol developments strategies in China. Although, several AF biocontrol developments studies are reported our focus is only the methods which utilizes atoxigenic A.flavus as a biocontrol agents. Hence, I made some corrections as section “AF biocontrol developments in China” is subtitle under section “Atoxigenic A. flavus as AF biocontrol agents”.

NB: All corrected sections in the manuscript are indicated in green highlight

Thanks again for commenting our manuscript

Reviewer 2 Report

The manuscript is an excellent review of the aflatoxin problem in China, and of most recent developments done in there concerning this research topic. The manuscript is very well written and the English is good. All the references cited are pertinent and comprehensive, providing a very exhaustive overview of Chinese research work and state-of-art in the aflatoxins topic. I attach the manuscript with some minor comments and suggestions of corrections to authors improve a little more the excellence of their work. Please try to correct some missing spaces throughout the text.

Author Response

Reviewer 2

Thank you! We found your comments extremely helpful and have revised accordingly. Our responses are given in a point -by-point manner below. Changes to the manuscript are shown by a yellow  highlight.

Comments:

The manuscript is an excellent review of the aflatoxin problem in China, and of most recent developments done in there concerning this research topic. The manuscript is very well written and the English is good. All the references cited are pertinent and comprehensive, providing a very exhaustive overview of Chinese research work and state-of-art in the aflatoxins topic. I attach the manuscript with some minor comments and suggestions of corrections to authors improve a little more the excellence of their work. Please try to correct some missing spaces throughout the text.

Response: I have corrected the whole manuscript according your valuable comments. All changes made in the text are shown in yellow highlights. 

Thanks again for commenting our manuscript

Reviewer 3 Report

The work "Distribution of Aspergillus fungi and recent aflatoxin reports, health risks, and advances in developments of biological mitigation strategies in China" presented to me for review is a very valuable scientific work. I appreciate the great contribution of the work to the preparation of the manuscript and the search for numerous and valuable scientific literature cited in the work. The work has a very good and logical layout of chapters. It deals with many aspects in the field of the harmfulness of Aspergillus fungi and the aflatoxin produced by them. As a reviewer, I have a request to expand the chapter "2.4. Animal feed and dairy products". It would be good to develop the harmfulness of aflatoxin in relation to animals, especially since historically the first harmful effects of this mycotoxin were found in turkeys. So I recommend expanding on this topic. Write down the risk of aflatoxin in animals, especially in the context of animals raised in China. Is there high exposure of people who consume milk and other animal products to aflatoxin metabolites? Which feed products intended for animal nutrition pose the greatest risks. It would also be worth adding two sentences on this topic to the summary of the work.

Author Response

Thank you! We found your comments extremely helpful and have revised accordingly. Our responses are given in a point -by-point manner below. Changes to the manuscript are shown by a yellow  highlight.

Reviewers 3

The work "Distribution of Aspergillus fungi and recent aflatoxin reports, health risks, and advances in developments of biological mitigation strategies in China" presented to me for review is a very valuable scientific work. I appreciate the great contribution of the work to the preparation of the manuscript and the search for numerous and valuable scientific literature cited in the work. The work has a very good and logical layout of chapters. It deals with many aspects in the field of the harmfulness of Aspergillus fungi and the aflatoxin produced by them. As a reviewer, I have a request to expand the chapter "2.4. Animal feed and dairy products". It would be good to develop the harmfulness of aflatoxin in relation to animals, especially since historically the first harmful effects of this mycotoxin were found in turkeys. So I recommend expanding on this topic. Write down the risk of aflatoxin in animals, especially in the context of animals raised in China. Is there high exposure of people who consume milk and other animal products to aflatoxin metabolites? Which feed products intended for animal nutrition pose the greatest risks. It would also be worth adding two sentences on this topic to the summary of the

Point -by -point responses to comments by reviewers -3

Thanks for your valuable comments

Comment: "2.4. Animal feed and dairy products". It would be good to develop the harmfulness of aflatoxin in relation to animals, especially since historically the first harmful effects of this mycotoxin were found in turkeys. So I recommend expanding on this topic”

As you have mentioned clearly, aflatoxins were discovered at early 1960s, when they were identified as causative agents of “turkey X” disease, due to peanut bases feeds originated from south America. As of your comments, several harmful effects of aflatoxin in animals have been well studies in different countries across different spices.  As to the scope of this manuscript we have incorporated the general impacts of aflatoxins on animals.

Comment: Is there high exposure of people who consume milk and other animal products to aflatoxin metabolites?

Response: Section “4. Health impacts of AFs in China “is meant to describe the human health impacts of aflatoxins due to dietary exposures.  As to your suggestions it is obvious that livestock products originated from aflatoxin pre-exposed animals might have harmful health impacts to the consumers. Several authors have reported the impaired growth, and other chronic health problems in children which their main food sources are animal products (Eg. milk).   Unfortunately, this issue specific to China is not well reported.

Comment : Which feed products intended for animal nutrition pose the greatest risks?

Thanks again for your nice comments. Your insightful comments are well appreciated. 

Disregarding of proper handling, processing and storage, nutritious foods are highly susceptible to fungal growth as well as aflatoxin contamination. Similarly, it is expected for nutritious feeds types are susceptible for fungal proliferation as well as aflatoxin contaminations. Thus, specific feeds types should be of our focus. Unfortunately, due to the scope of the manuscript we did not consider the issues in this manuscript.  However, we have tried to address your valuable comments in the manuscript. Like any food commodities, unless, properly harvested, dried, processed and stored any feed types could also be suitable for fungal proliferation as well as aflatoxin contamination.

Change: following your suggestion, the following paragraph has been included in the manuscript.

“The aflatoxins were discovered at early 1960s, when they were identified as causative agents of “turkey X” disease, due to peanut bases feeds originated from south America [81]. Onwards, several researches have been reported as long-term exposure of animals to sub acutely toxic levels of AFs is reported to cause several animal health implications such as feed refusal, increased disease susceptibility, weight loss, inferior egg shell, reduced milk yield, defects on carcass quality, reproduction defects [82,83], and other severe impacts like liver lesions/tumors, hepatotoxicity, nephrotoxicity, gastroenteritis, and teratogenicity [84,85].

Hence, like any crop commodities targeted for human consumption, animal feeds (Eg. Silages) should not have higher fungal loads beyond the recommended limits (103-104 cfu/g) [86] and should not be exposed to AF contaminations. Like any food commodities, unless, properly harvested, dried, processed and stored any feed types could also be suitable for fungal proliferation as well as aflatoxin contamination.”   

Thanks again for commenting our manuscript